# Role of Perceived Social Support on the Association between Physical Disability and Symptoms of Depression in Senior Citizens of Pakistan

**DOI:** 10.3390/ijerph17051485

**Published:** 2020-02-25

**Authors:** Azam Tariq, Tian Beihai, Nadeem Abbas, Sajjad Ali, Wang Yao, Muhammad Imran

**Affiliations:** 1Department of Sociology, College of Humanities and Social Sciences, Huazhong Agricultural University, Wuhan 430070, China; azam_tariq@webmail.hzau.edu.cn; 2Institute of Social & Cultural Studies, University of the Punjab, Lahore 54000, Pakistan; nadeem544abbas@gmail.com; 3College of Economics and Management, Huazhong Agricultural University, Wuhan 430070, China; sajjad@webmail.hzau.edu.cn; 4Department of Social Security, College of Humanities and Social Sciences, Huazhong Agricultural University, Wuhan 430070, China; 18627064001@163.com; 5Department of Computer Science and Engineering, Shanghai Jiao Tong University, Shanghai 200240, China; Muhammad_imran@sjtu.edu.cn

**Keywords:** perceived social support, mediator, symptoms of depression, physical disability, senior citizens

## Abstract

An emerging body of literature has implied that perceived social support is known as an upstream element of cognitive health. Various dimensions of perceived social support may have divergent influence on physical and cognitive health in later life. The present study aimed to investigate the mediating role of perceived social support on the relationship between physical disability and symptoms of depression in senior citizens of Pakistan. The data were collected from three metropolitan cities (Lahore, Faisalabad, Multan) in the Punjab province of Pakistan and 100 participants were approached from each city with a total sample size of 300. The results demonstrated that family support, friends’ support, and significant others’ support mediated the association between physical disability and symptoms of depression, with an indirect effect of 0.024, 0.058, and 0.034, respectively. The total direct and indirect effect was 0.493. Physical disability was directly associated with symptoms of depression and greater physical disability predicted a higher level of symptoms of depression. Perceived social support, including family support, friends’ support, and significant others’ support, showed an indirect association with symptoms of depression. Furthermore, family support and friends’ support were more significantly associated with symptoms of depression as compared to significant others’ support. The research discoveries have better implications for health care professionals, hospice care workers, and policy makers. A holistic approach is required to prevent senior citizens from late-life mental disorders.

## 1. Introduction

According to the United Nation, the global population over 60 years of age was 962 million in 2017 and is expected to be double to 2.1 billion by the year 2050. Two-thirds of the total older population live in developing countries. The rapid projections show that by the year 2050, almost 8 of 10 elderly people will be residents of developing regions [1]. Pakistan is a developing country and the population aged 60 or over was 11.3 million in 2017, and it is projected to increase to more than 43.3 million in 2050, which will account for nearly 16% of the overall population [2].

This rapid demographic transition has led to increased complicated and interconnected problems, including socioeconomic, physical, social, and cognitive health problems of elderly people, and previous studies discovered that depression in the elderly has become a global public health problem [3,4,5]. A research revealed 16.52% life-time prevalence of depression in elderly people. However, the prevalence of depression varied among countries [6]. In Pakistan, the prevalence of depression in senior citizens is rising day by day, and many studies have reported 18% to 66% in both rural and urban settings [7,8,9,10]. It is expected that by the year 2020, depression will be the leading burden of disease in the world [11]. Depression is interrelated with adverse socioeconomic outcomes, like emotional and cognitive suffering, poverty, family conflicts, rise in health care expenditure, and death rate [12].

Physical disability is defined as impairments or limitations in daily activities and restricted social involvement [13]. The inability to perform daily tasks of self-care and limitations in an individual’s capacity to participate in the social and physical environment is also known as physical disability [14]. The decline in physical health is mostly associated with aging and acts as a big stressor in later life [15,16]. It has been proven that failure to carry out activities of daily living (ADL) and instrumental activities of daily living (IADL) are associated with depression in older people [17,18]. Physical or functional disability often leaves elderly people vulnerable to depression [19,20,21,22]. It has been found that an increased level of ADL predicts a lower level of depression [23]. A cross-sectional study carried out in China with a sample size of 372 elderly people found that less dependency in activities of daily living was associated with lower level of depression [24]. Furthermore, some longitudinal studies support the argument that it leads to higher symptoms in older adults [25]. The given influence of physical disability on depression highlights the need to find directions to deal with it. 

Social support is one of the factors which influences symptoms of depression in elderly people. Social support refers to the series of accessible support to a person through their social relationships with other people [26]. It also includes information or knowledge, emotional aid, substantial help, and self-sufficiency that individuals gain through mutual relationships [27]. Perceived social support, also known as subjective support, is defined as the level of satisfaction of being empathized, valued, and supported in the society [28]. It represents how much a person feels safe and companionable [29]. Social support has received sufficient attention because of its function of minimizing stress and mental health problems and preventing the harmful effects of physical disability on psychological well-being [30]. Perceived social support plays the role of a powerful stress reducer and has been found to be effective in minimizing or mediating the association between physical disability and symptoms of depression [31,32]. A study carried out on Turkish older people with 102 respondents discovered a significant effect of perceived social support on depression and found that lower perceived social support was associated with greater depression [29]. Building up this association, a study found that subjective support mediates the relationship between ADL and symptoms of depression in elderly [24].

Perceived social support can be further divided into three dimensions, as family support, friends’ support, and significant others’ support [33]. According to the best of our knowledge, there have been no studies done to discover the effects of family support, friends’ support, and significant others’ support on the relationship between physical disability and depression in senior citizens (people aged 60 or above).

According to the stress-buffering hypothesis, social support has positive effects on health and well-being by protecting individuals from the harmful effects of stressors. Social support is also considered as a copping source and physical disability as a stressor [34]. So, it is mandatory to find out what dimensions of perceived social support are linked with physical disability and symptoms of depression.

The primary objective of the present study is to discover whether perceived social support mediates the association between physical disability and symptoms of depression in senior citizens living in urban areas of the Islamic Republic of Pakistan.

The current study first hypothesizes that physical disability is significantly associated with greater symptoms of depression (H1). Then, three dimensions of perceived social support could mediate the association between physical disability (ADL, IADL) and symptoms of depression (H2). This study further hypothesizes that family support is associated with symptoms of depression more significantly than friends’ support and significant others’ support (H3), as mentioned in the hypothetical model (Figure 1). The discoveries of this investigation will highlight the role of perceived social support on physical disability and symptoms of depression and is expected to be helpful in developing policies to promote perceived social support for the well-being of senior citizens.

## 2. Materials and Methods 

### 2.1. Study Participants 

A cross-sectional design study was carried out over four months in 2018 with a sample size of 300 senior citizen participants from three big cities, including provincial capital city Lahore, Faisalabad (industrial city), and Multan, which is the historical city and also known as a city of saints in the Punjab province of Pakistan (Figure 2). The participants were approached through a purposive sampling technique to perform an interview-based questionnaire with the help of three research assistants with a relevant field of study. Participants aged 60 years or over (the age mentioned in Pakistan government senior citizen act and referred by United Nations Organization), with a willingness to participate on a voluntary basis and ability to communicate in Urdu (national language of Pakistan) were included in the study. Senior citizens with severe hearing and sight impairment, severe physical injuries, and terminal illness were excluded from the study. 

All senior citizens who were willing to participate in this investigation were informed about the objectives of conducting research. The personal and private information was ensured to be kept secret and aggregate data were used.

### 2.2. Measurement Tools

#### 2.2.1. Symptoms of Depression

The primary dependent outcome variable “symptoms of depression” in senior citizens was assessed via the Beck depression inventory (BDI) scale [35]. The BDI scale is a standardized and self-reported measurement tool that consists of 21 items ranging on a four-point scale, 0 to 3, and a higher score of total 63 scores indicates a higher level of symptoms of depression. It is a widely used scale for both clinical and research purposes and measures the presence of cognitive, vegetative, psychomotor, and motivational features of depression [7]. The BDI scale measures the mood of subjects for the previous two weeks [29]. The scale has been translated into Urdu language (Pakistan national language) and has been used in previous research in the same setting [7]. The Cronbach alpha for this sample was 0.86, which indicates its reliability.

#### 2.2.2. Physical Disability

Physical disability was assessed through activities of daily living (ADL), including its subscales [36], and instrumental activities of daily living (IADL) with subscales [37]. The ADL subscale measures six types of abilities, including taking baths, feeding, dressing, toileting, transferring, and continence. The IADL subscale is used to measure eight types of complex activities, like telephone use, going shopping, transportation, finance handling, laundry, taking medicine, food preparation, and housekeeping. The scores range from 1 to 4 and total scores are 56. The higher obtained scores show greater physical disability. The Cronbach alpha for this study was 0.85.

#### 2.2.3. Perceived Social Support

Perceived social support was measured by using a multidimensional scale of perceived social support (MSPSS) [38]. This scale measures the overall score of perceived social support including three subscales, Family support, Friends support and significant others support. This scale has been translated in Urdu. Furthermore, this scale has been used in previous studies showing good reliability and psychometric properties [39,40]. The Cronbach alpha for the current sample was 0.81. 

### 2.3. Analytical Techniques 

The data analysis was carried out through the statistical package for social sciences (IBM SPSS-21, IBM Corp., Armonk, N.Y., USA). Prior to executing key analysis, descriptive statistics were calculated on sociodemographic characteristics to uncover the distribution of different variables and to determine proportions for the prevalence of depression. The range, mean (X), and standard deviation (SD) were found for the perceived social support, including its subscales (family support, friends’ support, significant others’ support), physical disability, and symptoms of depression. Continuous variables were evaluated through one-way analysis of variance with the Student–Newman–Keuls test for post hoc multiple comparisons. The correlations between physical disability, perceived social support, including its dimensions (family support, friends’ support, and significant others’ support), and symptoms of depression were determined.

Multiple linear regression was then executed to determine the contribution of physical disability and perceived social support (family support, friends’ support, significant others’ support) on symptoms of depression. The symptoms of depression were kept as a dependent variable in this analysis. PROCESS model-4 [41] was used to discover the mediation effect of three dimensions of perceived social support on the association between physical disability and symptoms of depression, with symptoms of depression as an outcome variable. 

## 3. Results

The socio-demographic characteristics and symptoms of depression for the current sample are presented in Table 1. The majority of the research participants were male (72%), the age of the respondents ranged from 60 to 90 years, among them, 51.6% were between 60 and 69, 33.6% were between 70 and 79 years, and the remaining 14.6% were ≥80 years of age. A total of 74.9% of the respondents were at the education level of high school or less, whereas almost 25% of the respondents had education above high school level; 55.6% participants belonged to a joint family system, 38.6% respondents had a family size of one to four members, whereas 48.6% had five–eight, and 12.6% had a family size of ≥eight members; 64.3% were married, 39.3% were household head, 47.3% of the respondents had family as a source of income, 44% participants belong to the family with two earning hands; 41.3% had less than 30,000 PKR (Pakistan Rupees) and the remaining respondents had more than 30,000 PKR monthly income; 27.6% of the respondents were living at a distance less than 3 Km from hospital, however, the majority of the respondents (52.6%) were living at the distance of 4–7 km from the hospital; 14.6% of the respondents were not suffering from any chronic disease, whereas 64.2% had at least one or two chronic diseases, including asthma, diabetes, and hypertension. Moreover, 60% of the participants were living with a spouse, 21.3% of them were disabled by birth or because of an accident, a separate room at home was available for 52% of the elderly, and 62.6% of the respondents considered their children as future security.

In results, we compared the socio-demographic variables, including gender, age, level of education, family system, family size, marital status, respondents current status, current source of income, total earning hands in family, monthly household income, distance from hospital, number of chronic diseases, living status, physical disability, smoking habit, separate room at home, and children as future security, with the criterion variable symptoms of depression. It was found that the family system, respondent’s household status, total earning hands, distance from the hospital, number of chronic diseases, smoking habit, and having a separate room at home did not show significant association with symptoms of depression. 

Post hoc multiple comparisons found that female respondents, elderly above the age of 70 years, and the respondents with a high school or lower level of education scored significantly higher on the symptoms of depression scale. The respondents with a family size of one to four, single/divorced/widowed status, and with family as the current source of income had significantly higher symptoms of depression. Furthermore, the respondents whose monthly income was less than 30,000 PKR, living with others (children/relatives), and interestingly, non-disabled, scored higher on symptoms of depression. Lastly, the participants who considered their children as future security and did not have a separate room at home scored higher symptoms of depression.

Table 2 shows the Pearson correlation coefficient analysis among physical disability, perceived social support, family support, friends’ support, significant others’ support, and symptoms of depression. Physical disability and symptoms of depression were directly correlated (r = 0.469, *p* < 0.01), signifying that greater physical disability in the elderly was directly associated with a higher level of symptoms of depression, whereas perceived social support, family support, friends’ support, and significant others’ support were indirectly associated with symptoms of depression (r = −0.411, −0.344, −0.343, −0.379, *p* < 0.01), indicating that a higher level of perceived social support, family support, friends’ support, and significant others’ support was associated with a lower level of symptoms of depression. This study found an indirect correlation between perceived social support and physical disability (r = −0.432, *p* < 0.01). Likewise, three dimensions of perceived social support (family support, friends’ support, significant others’ support) were inversely correlated with physical disability (r = −0.177, −0.275, −0.263, *p* < 0.01), suggesting that greater physical disability was associated with a lower level of family support, friends’ support, and significant others’ support.

The socio-demographic variables with a significant influence on symptoms of depression were involved in a multiple linear regression model. Hence, the gender, age, level of education, marital status, who they were living with, family size, current source of income, average monthly income, physical disability (birth/accidental), and children as future security were considered confounding variables and were controlled in the current analysis. Prior to running multiple linear regression, it was ensured that there was no violation of assumptions like normality, linearity, and multicollinearity. The results (R^2^ = 0.452, F = 10.935, *p* < 0.01) show the significance of the regression model. A direct association was found between physical disability and symptoms of depression (β = 0.244, *p* < 0.01), as predicted in H1, and a lower level of physical disability was associated with a lower level of symptoms of depression in elderly people. The results demonstrate that family support (β = −0.152, *p* < 0.01), friends’ support (β = −0.136, *p* < 0.01), and significant others’ support (β = −0.120, *p* < 0.05) were inversely associated with symptoms of depression among older people. Moreover, family support (β = −0.152) had more negative association with symptoms of depression as compared to friends’ support (β = −0.136), and significant others’ support (β = −0.120) (Table 3).

The mediating effect of three dimensions of perceived social support (family support, friends’ support, significant others’ support) on the association between physical disability and symptoms of depression is mentioned in Table 4. PROCESS macro model-4 (v3.3) (http://www.processmacro.org/download.html) was used to analyze the direct and indirect effect of three dimensions of perceived social support and physical disability [41]. This analysis was performed through bootstrapping strategy with 5000 resamples. Status of residence and average monthly income were used as covariates. 

The results indicate the significance of overall models (R^2^ = 0.273, 0.379, F = 37.207, 29.902, *p* < 0.01). It was found that physical disability was strongly associated with symptoms of depression (B = 0.495, *p* < 0.01) before entering perceived social support into the equation, whereas this effect was then mediated to (B = 0.377, *p* < 0.01) by family support (B = −0.296, *p* < 0.01), friends’ support (B = −0.287, *p* < 0.01), and significant others’ support (B = −0.331, *p* < 0.01). The indirect effect of family support, friends’ support, and significant others’ support was 0.024 (−0.084 × −0.296), 0.058 (−0.203 × −0.287), 0.034 (−0.103 × −0.331), respectively. The total direct and indirect effect was 0.493 (0.377 + 0.024 + 0.058 + 0.034). The results demonstrate, as hypothesized in H2, that three dimensions of perceived social support (family support, friends’ support, significant others’ support) play mediating roles on the association between physical disability and symptoms of depression in senior citizens (Figure 3). 

## 4. Discussion

The present research concurrently discovered the associations among perceived social support, physical disability, and symptoms of depression and examined the mediating effect of family support, friends’ support, and significant others’ support on the association between physical disability and symptoms of depression in senior citizens. Control variables like the status of residence and average monthly income, if not included, could perhaps confuse the associations between perceived social support, physical disability, and symptoms of depression. 

The sociodemographic characteristics that were significantly associated and showed higher levels of symptoms of depression indicate the secure social status, higher self-adjustment, greater adaptability to the social situations and the proper utilization of available social support. Hence, unavailability of the same leads to greater mental distress [42]. The risks of late-life depression may be minimized by improving social relationships, strengthening ties, and most importantly, enhancing the utilization of existing social support rather than stipulating other resources of support [24].

The current study discovered a significant direct association between physical disability and symptoms of depression by confirming the previous research findings of the association between limitations in activities and socio-psychological problems [17,24]. A study done in Pakistan on rural elderly people which included 146 respondents found a direct relationship between physical disability and depression [43]. Furthermore, a study found similar results that lower ADL (activities of daily living) functioning was associated with a greater level of depression in the elderly [29], and it has a negative impact on cognitive health and is a fundamental cause of socio-psychological problems in older people [19,21,22,44].

The three dimensions of perceived social support (family support, friends’ support, and significant others’ support) were found to be inversely associated with symptoms of depression. A study carried on Turkish elderly with 102 participants found that higher perceived social support predicted lower level of depression [29]. Furthermore, subjective support (perceived social support) showed an inverse relationship between activities of daily living and depressive symptoms in Chinese elderly [24]. In addition, our findings showed that family support and friends’ support have more significant associations with symptoms of depression as compared to significant others’ support. Citizen welfare trust conducted research and discovered that almost 98% of the senior citizens of Pakistan favor living with family rather than staying at old age homes or somewhere else [45]. Sufficient literature is available suggesting that family support and informal friend support are the primary sources of emotional support for the elderly [46,47,48]. Moreover, it was found that family support and friends’ support is important for elderly people of Asia [49,50,51]. Conventionally, elderly people mostly seek care and support from the family members [52].

The findings of the mediation effect model show that family support, friends’ support, and significant others’ support mediated the effect on the relationship between physical disability and symptoms of depression. Lower physical disability was significantly associated with a lower level of symptoms of depression. A study conducted on 372 Chinese elderly found that perceived social support and utilization of support mediated the relationship between impairment of ADL and depression [24]. Our research contributes to existing literature with in-depth discoveries and has imperative inferences for geriatric professionals, care providers, and policy interventions, as perceived social support is more acquiescent to cope. Previous studies focused on different dimensions of social support and mostly checked the direct effect, but the recent investigation is unique on the basis of its objectives and methodology. Our findings suggest that in the absence of any of the three dimensions of perceived social support, senior citizens with physical disability have a greater risk of suffering from symptoms of depression in later years of life, and emphasize the significance of the perceived support from family, friends, and significant others to prevent physical and cognitive health obstacles in late life.

The current research has strengths and some limitations. According to our knowledge, based on the existing literature, this is the first-ever study, especially for the Pakistan context, which has investigated the role of perceived social support on the association between physical disability and depression in senior citizens of three metropolitan cities of the Punjab province of Pakistan. This research filled the literature gap from a different point of view, by finding the influence of physical disability on symptoms of depression rather than finding the influence of symptoms of depression on physical disability, as in previous studies. In limitations, the data were collected from three different cities, which may affect the generalizability of the findings. Furthermore, the sample size of the study may be considered as a limitation, and the cross-sectional approach can affect the direct casual evidence; thus, further studies based on the longitudinal approach and with a larger sample size are recommended to discover causality.

## 5. Conclusions

The current study attempted to discover the role of perceived social support (family support, friends’ support, significant others’ support) as a mediator on the association between physical disability and symptoms of depression in senior citizens of Pakistan. The major discoveries in this research may not be surprising, because Pakistan’s culture is based on collectivism, as a collectivist culture refers to the structure of society in which individuals give priority to commitment, conformity, and group loyalty, and possession of a sense of belonging rather staying in isolation [53]. Currently, rapid demographic change to individualism and the nuclear family system has led senior citizens to suffer from social isolation, with a lack of perceived social support. Our findings suggest that all three dimensions, family support, friends’ support, and significant others’ support, of perceived social support are important to mediate the association between physical disability and symptoms of depression. The findings facilitate the evidence that perceived social support, including its dimensions, has passive consequences on physical limitations and symptoms of depression among senior citizens. Furthermore, it was found that a lower level of perceived social support leads to greater symptoms of depression.

In short, the major discoveries revealed that perceived social support and its dimensions are negatively correlated with symptoms of depression, while physical disability showed a direct association with symptoms of depression and the three dimensions of perceived social support play the role of mediator on the association between physical disability and symptoms of depression. The findings reveal the difficulties in the lives faced by senior citizens. Developing intervention policies and programs to safeguard senior citizens from debilitating effects in the absence of perceived social support and providing both material and nonmaterial aid for independent living to diminish the vulnerability to physical and cognitive health disorders in senior citizens are highly needed to facilitate them with a better quality of life in their late years of life.

## Figures and Tables

**Figure 1 ijerph-17-01485-f001:**
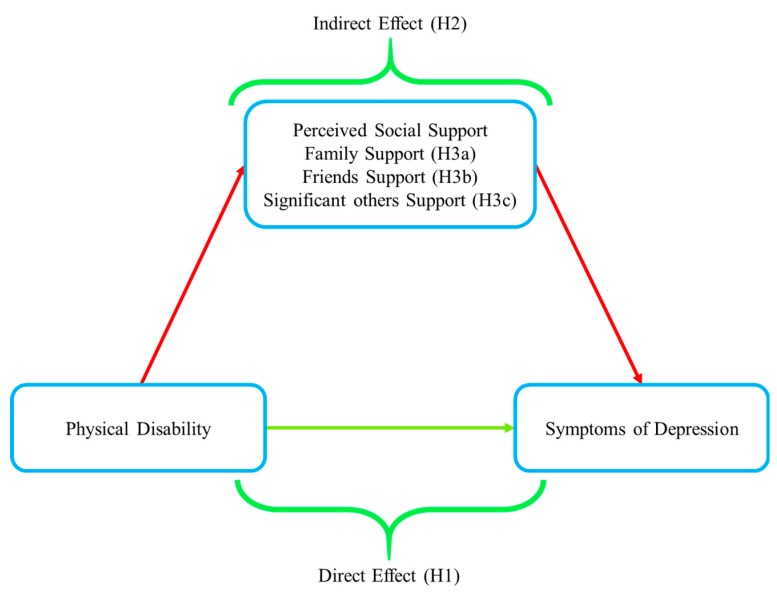
Hypothetical model.

**Figure 2 ijerph-17-01485-f002:**
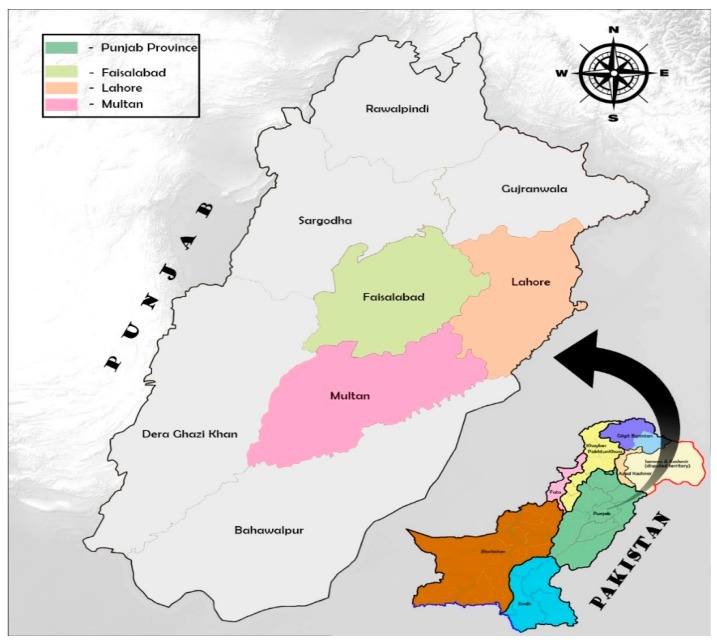
Survey map of selected cities in Punjab province, Pakistan (source: authors’ own).

**Figure 3 ijerph-17-01485-f003:**
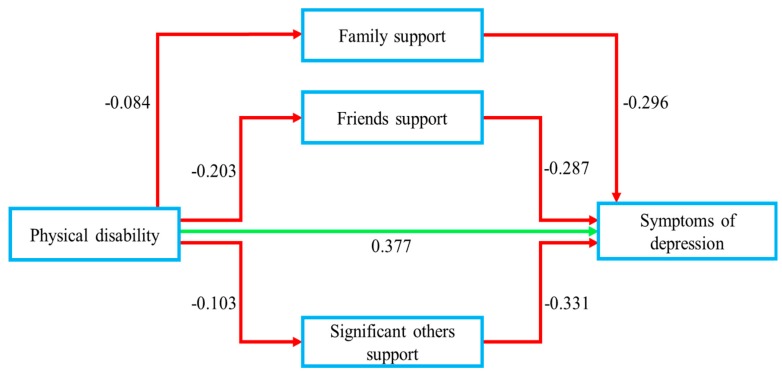
Mediating effect of perceived social support affecting physical disability and symptoms of depression among senior citizens of Pakistan (*N* = 300).

**Table 1 ijerph-17-01485-t001:** Comparison of different socio-demographic characteristics with depressive symptoms.

Socio-Demographic Variables	No.	Percentage (%)	Symptoms of Depression $\bar{X}$ ± *SD*	*F*	*p*
Gender				6.286	0.013
Male	216	72	33.14 ± 8.67		
Female	84	28	30.33 ± 8.82		
Age (Years)				25.928	0.000
60–69	155	51.6	29.58 ±8.72		
70–79	101	33.6	33.60 ± 7.70		
80 years or above	44	14.6	39.25 ± 6.85		
Education				10.972	0.000
Primary school or less	58	19.3	37.08 ± 7.03		
Middle school	88	29.3	34.30 ± 8.32		
High school	79	26.3	29.07 ± 7.94		
Intermediate school	41	13.6	29.14 ± 10.20		
Bachelor’s or above	34	11.3	30.73 ± 8.12		
Family Structure				1.221	0.270
Joint	167	55.6	31.85 ± 8.55		
Nuclear	133	44.3	32.98 ± 9.07		
Family Size				3.930	0.021
1–4	116	38.6	32.59 ± 8.95		
5–8	146	48.6	33.11 ± 8.75		
Above 8	38	12.6	28.71 ± 7.68		
Marital Status				7.749	0.006
Married	193	64.3	31.31 ± 8.82		
Single/Divorced/Widowed	107	35.6	34.23 ± 8.45		
Respondent’s Status				3.175	0.076
Is Household head	118	39.3	31.23 ± 8.76		
Is not Household head	182	60.6	33.08 ± 8.76		
Current Source of Income				6.251	0.000
Salary	6459	21.3	29.93 ± 8.25		
Pension	35	19.6	29.67 ± 8.29		
Agriculture/Property	142	11.6	33.34 ± 8.40		
Family		47.3	34.31 ± 8.85		
Earning Hands				0.783	0.458
1	91	20.3	31.42 ± 8.99		
2	132	44	32.91 ± 8.15		
3	77	25.6	32.49 ± 9.59		
Monthly Income				9.144	0.003
˂30,000	124	41.3	34.16 ± 8.06		
≥30,000	176	58.6	31.08 ± 9.07		
Distance from Hospital				1.218	0.297
˂3 km	83	27.6	31.71 ± 9.03		
4–7 km	158	52.6	33.09 ± 8.88		
≥8 km	59	19.6	31.28 ± 8.14		
No. of Chronic diseases				1.489	0.218
0	44	14.6	31.09 ± 9.86		
1	110	36.6	33.29 ± 7.69		
2	83	27.6	32.95 ± 9.14		
3	63	21	30.82 ± 9.21		
Living status				47.916	0.000
Living with Spouse	180	60	29.68 ± 8.96		
Living with other’s (Children/Relatives)	120	40	36.35 ± 6.81		
Physical disability (Birth/accidental)				6.461	0.012
Disabled	64	21.3	34.81 ± 8.31		
Non-disabled	236	78.6	31.69 ± 8.81		
Smoking habit				0.028	0.867
Smoker’s	130	43.3	32.45 ± 8.68		
Non smoker’s	170	56.6	32.28 ± 8.90		
Children as a future security				23.743	0.000
Yes	188	62.6	30.51 ± 8.75		
No	112	37.3	35.44 ± 7.98		
Separate room at home				3.059	0.081
Available	156	52	31.50 ± 9.50		
Not available	144	48	33.27 ± 7.87		

**Table 2 ijerph-17-01485-t002:** Descriptive statistics and Pearson correlation analysis.

Variables	Range	Mean	SD	Perceived Social Support	Family Support	Friends’ Support	Significant Others’ Support	Symptoms of Depression
Physical disability (ADL, IADL)	22–52	35.97	6.61	−0.432 **	−0.177 **	−0.275 **	−0.263 **	0.469 **
Perceived social support	21–71	43.44	10.03	-	0.699 **	0.472 **	0.667 **	−0.411 **
Family support	4–26	15.14	4.53	-	-	0.231 **	0.480 **	−0.344 **
Friends’ support	4–26	13.03	5.34	-	-	-	0.215 **	−0.343 **
Significant others’ support	4–26	15.72	3.95	-	-	-	-	−0.379 **
Symptoms of Depression	15–57	32.35	8.79	-	-	-	-	-

SD = Standard deviation, ** = Correlation is significant at 0.01 level.

**Table 3 ijerph-17-01485-t003:** Multiple linear regression on symptoms of depression.

Dependent Variable: Symptoms of Depression	Unstandardized Coefficient	Standardized Coefficient
	B (SE)	Β
Gender	−2.503 (0.944)	−0.128 **
Age (Years) (Ref. 60-69)		
70–79	−0.430 (0.997)	−0.023
80 or above	2.899 (1.480)	0.117 *
Level of Education (Ref. Primary or less)		
Middle (8 Years)	0.977 (1.268)	0.051
High school (10 years)	−3.074 (1.313)	−0.154 *
Higher secondary school (12 years)	−2.749 (1.507)	−0.108
Bachelor’s or above	−1.207 (1.731)	−0.044
Marital status	−0.816 (0.968)	−0.045
Living with	1.516 (1.064)	0.085
Family size (Ref. 1–4 members)		
5–8	0.180 (0.918)	0.010
Above 8	−1.617 (1.418)	−0.057
Current source of income (Ref. Salary)		
Pension	1.082 (1.283)	0.049
Agriculture/property	2.191 (1.536)	0.080
Family	1.550 (1.183)	0.088
Average monthly income (PKR)	−1.478 (0.855)	−0.083
Physical disability (birth/accidental)	−1.205 (1.045)	−0.056
Children as future security	0.617 (0.915)	0.034
Physical disability (ADL, IADL)	0.325 (0.078)	0.244 **
Family support	−0.294 (0.102)	−0.152 **
Friends’ support	−0.224 (0.082)	−0.136 **
Significant others’ support	−0.268 (0.122)	−0.120 *
R^2^	0.452	
F	10.935 **	

Ref = Reference, SE = Standard Error, ** = *p* ˂ 0.01, * = *p* ˂ 0.05.

**Table 4 ijerph-17-01485-t004:** Mediation effect analysis based on PROCESS (Model 4).

Variables	B (SE)	LLCI	ULCI
Outcome variable: Symptoms of depression			
Physical disability	0.4952 ** (0.074)	0.3483	0.6421
Average monthly income	−2.7784 ** (0.883)	−4.5172	−1.0395
Status of residence	3.4804 ** (1.005)	1.5016	5.4591
R^2^	0.273		
F	37.207 **		
Outcome variable: Symptoms of depression			
Family support	−0.296 ** (0.102)	−0.498	−0.094
Friends’ support	−0.287 ** (0.080)	−0.445	−0.128
Significant others’ support	−0.331 ** (0.120)	−0.568	−0.093
Physical disability	0.377 ** (0.071)	0.236	0.518
Average monthly income	−2.621 ** (0.821)	−4.238	−1.004
Status of residence	2.518 ** (0.948)	0.652	4.385
R^2^	0.379		
F	29.902 **		

Ref = Reference, SE = Standard Error, LLCI = Lower level confidence interval, ULCI = Upper level confidence interval, ** = *p* ˂ 0.01.

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
