# Peer review of "Role of Perceived Social Support on the Association between Physical Disability and Symptoms of Depression in Senior Citizens of Pakistan"

_ijerph, 2020, doi:10.3390/ijerph17051485_

Round 1

Reviewer 1 Report

Congratulations on conducting a very relevant study. Your English needs some work. Overall, the study was very sound. I found the discussion section too long and redundant with the results section. Your results section should give us all the stats so you do not need to walk us back through each individual statistical finding in the discussion. Rather, the discussion should be a quick summary of the finds, followed by a discussion of the implications of the findings and limitations of the study. You started out on pg 11, line 335 mentioning the limitations and strengths, but you never state what the limitations are. I would strongly recommend shortening the discussion as I mentioned above and actually stating the limitations of the study.

Author Response

Point 1. Moderate English changes required.

Response;

Respected reviewer, we appreciate your comment. We have modified and made English changes in revised manuscript. Kindly check in track changes.

Point 2. I found the discussion section too long and redundant with the results section. Your results section should give us all the stats so you do not need to walk us back through each individual statistical finding in the discussion. Rather, the discussion should be a quick summary of the finds, followed by a discussion of the implications of the findings and limitations of the study.

Response;

Thanks for your valuable comments and suggestions. We have followed the given suggestions and modified our manuscript accordingly. Kindly check line No. 258-275 and line No. 3018-311. We have tried our best to summarize the discussion section.

Point 3. You started out on pg 11, line 335 mentioning the limitations and strengths, but you never state what the limitations are. I would strongly recommend shortening the discussion as I mentioned above and actually stating the limitations of the study.

Response;

We have shortened the discussion and mentioned the strengths and limitations properly in discussion section as per recommendations. Please consider line No. 315-325. In addition, we also have clearly mentioned the study limitations.

We hope you will find our revised manuscript up to the standard of the publication.

Thank you for your valuable time.

Regards.

Reviewer 2 Report

This is a well executed project in general. There are, however, several issues that need to be more adequately addressed. First of all, I've never heard of the term permanent depression (line 4 of page 2). So the author might want to reword it or elaborate on it a bit more. Second, the causational relationship/association between physical disability and depression appears to be uni-directional. I believe it's plausible that depression may also exacerbate the symptoms of physical disability through several pathways. Third, the data were collected from three cities in Indiana, so the audience of this paper might have doubts about the generalizability of the findings from this paper.

Author Response

Point 1. English language and style are fine/minor spell check required 

Response;

We are grateful for your comment. We have modified our manuscript and sort out all the minor grammatical and spell mistakes. 

 Point 2. I've never heard of the term permanent depression (line 4 of page 2). So, the author might want to reword it or elaborate on it a bit more.

Response;

Respected reviewer, Thanks for your valuable comments and we apologize for the mistake and inappropriate discerption. Now, we have modified it properly. Kindly check in Line No. 48-49.

Point 3. The causational relationship/association between physical disability and depression appears to be uni-directional. I believe it's plausible that depression may also exacerbate the symptoms of physical disability through several pathways.

Response;

We appreciate your comments. We understand your concern properly. Actually, our focus in this study was on the relationship between physical disability and symptoms of depression. we also have found the correlations between these two variables. Furthermore, we have explained this point in the strengths and the limitation section of the study that “This research filled the literature gap from a different point of view by finding the influence of physical disability on symptoms of depression rather than finding the influence of symptoms of depression on physical disability in previous studies”. Please see Line No. 318-321.

Point 4. The data were collected from three cities in Indiana, so the audience of this paper might have doubts about the generalizability of the findings from this paper.

Response;

Dear reviewer, we are thankful for you concern data collection method. Our focus in this investigation was the urban residents and we collected the data from three metropolitan areas of the same province. Please see Line No. 106-109. Furthermore, as per your suggestions, to make it clear, we have added the statement in the limitations that “The data was collected from three different cities which may affect the generalizability of the findings”. Please see Line No. 321-322.

We hope you will find our revised manuscript up to the standard of the publication.

Thank you for your valuable time.

Regards.
